# Density and Home Range of Cats in a Small Inhabited Mediterranean Island

**DOI:** 10.3390/ani14162288

**Published:** 2024-08-06

**Authors:** Sara Molina-Bernabeu, Germán M. López-Iborra

**Affiliations:** Departamento de Ecología, Universidad de Alicante, 03690 Alicante, Spain; saramolinaph@gmail.com

**Keywords:** free-roaming cats, cat density, cat home range, small islands, Tabarca, trap-neuter-return

## Abstract

**Simple Summary:**

Domestic cats have spread worldwide, and their populations on islands have a significant impact on biodiversity. Particularly on small inhabited islands of tourist importance, cats can reach high densities. To evaluate cat impacts and plan cat population management, it is essential to know their population size and spatial distribution. This study examines the cat population on the small island of Tabarca (40 ha), near the Spanish Mediterranean coast, which includes a small village. Tabarca is included in the Natura 2000 Network due to its environmental value and bird populations. The overall cat density is among the highest reported (308 cats/km^2^), varying between the urban area (1084 cats/km^2^) and the uninhabited scrubland area (27 cats/km^2^). The home ranges of urban cats are much smaller (average 0.38 ha or 1.25 ha, depending on the estimation method) than those of cats in the scrubland (average 9.53 ha). These findings indicate that the urban area is a source of cats that colonize the scrubland. Despite the majority of cats being sterilized by the study’s end (89.5% of males and 91.7% of females), the population decline will be slow, taking many years to reach acceptable levels. Therefore, additional management measures are recommended to mitigate the cat population’s impact on biodiversity.

**Abstract:**

There is growing concern about effectively controlling cat populations due to their impact on biodiversity, especially on islands. To plan this management, it is essential to know the cat population size, sterilization rates, and space they use. Small inhabited islands can have very high cat densities; thus, this study aimed to evaluate cat density and home range on a small tourist island in the Spanish Mediterranean. Surveys in the urban area identified individual cats using a photographic catalog, and camera trapping was conducted in the scrubland area. GPS devices were fitted on three urban cats. The overall cat density was estimated to be 308 cats/km^2^, varying between the urban area (1084 cats/km^2^) and the uninhabited scrubland (27 cats/km^2^). Urban cats had smaller average home ranges (0.38 ha or 1.25 ha, depending on the estimation method) compared to scrubland cats (9.53 ha). Penetration of scrubland cats into the urban area was not detected. These results indicate that the urban area acts as a source of cats for the scrubland. Although the total sterilization rate was high (90.3%), the large cat population implies that the density would take over a decade to decrease to acceptable levels. Therefore, complementary measures for managing this cat population are recommended.

## 1. Introduction

The domestic cat (*Felis catus*) is one of the most widespread and abundant carnivores in the world [1]. They have dispersed worldwide with the help of humans, reaching all continents [2,3,4,5] and also islands [6,7,8]. Their generalist predatory behavior [9,10,11] causes them to prey on other species of mammals, reptiles, birds, and insects [12] in both urban and suburban habitats, as well as in uninhabited areas [2], posing a significant challenge for biodiversity conservation worldwide [13,14,15].

It is known that domestic cats exist in high densities in urban areas [16,17,18,19], which are also an important habitat for wild species [20,21]. The abnormally high presence of cats in urban areas and their feeding by humans in public places prevents their populations from fluctuating according to prey densities. This means that cats have the potential to drive prey species populations to extinction [22]. In many areas, the density of free-roaming cats exceeds that of other urban carnivores, such as the fox (*Vulpes vulpes*) and the badger (*Meles meles*). One study estimated that a territory that would normally be occupied by one or two pairs of native predators could support up to 35 cats in a peri-urban area [23]. Therefore, the impacts they have on urban wildlife are often greater than the impacts that native predators may generate [24].

In island environments, the domestic cat is one of the most widely introduced predators [8,25]. The presence of cats in these environments causes severe disturbances. These habitats are often more vulnerable to biological invasion, as the species that inhabit them are more susceptible to non-native predators, competitors, pathogens, and parasites [7,26,27,28]. The cat population on islands with small human populations can reach very high densities, as occurs on small Japanese islands [29]. Cats were originally introduced to islands to control rodents, but their diet also includes native bird species, especially during certain times of the year [30].

The generalization of the TNR (Trap-Neuter-Return) method for the ethical management of feral cat colonies has been proposed, but there is little rigorous evidence supporting the reduction of cat populations with this method [31,32,33]. Some studies have indicated that this method can lead to an increase in individuals in the colonies due to food provision, illegal abandonment of cats, reproduction, immigration, and/or attraction of cats from surrounding areas [34,35]. On some islands, the extraction and lethal control of stray cats is an effective conservation tool and is supported by conservation organizations [36]. However, social factors may hinder its implementation on inhabited islands [34,37]. Therefore, other management strategies, such as TNR, adoption programs, or the creation of sanctuaries, are more frequently accepted [38]. However, the TNR does not address critical issues such as predation by cats, especially in natural areas adjacent to feral cat colonies, risks related to zoonotic diseases and wildlife, public health, or the welfare of feral cats [35,39,40,41]. Another proposed mechanism to reduce potential cat predation involves establishing buffer zones around areas of high conservation value. Within these zones, housing development would be restricted, or ownership of domestic cats would be prohibited for residents living within a certain distance of the protected area [42,43,44].

Effective management plans for feral cat colonies must be based on a solid understanding of the community cat population and should be socially acceptable, promote feline welfare and public health, and protect wildlife. The first step in this approach is to obtain accurate estimates of cat population size and the factors that influence its dynamics. These factors include human ownership patterns [16], access to food [45,46], veterinary care [47], predator impact [23], and breeding opportunities [47]. Furthermore, maps of population density across urban and rural areas are vital for allocating scarce resources and guiding optimal interventions [38,48].

The small island of Tabarca is the only inhabited island in the Valencian Community and is included in the Natura 2000 Network, designated as a Special Protection Area (SPA, ES0000214) and Site of Community Importance (SCI, ES5213024). In addition to nesting bird species, it serves as a stopover point for migratory birds crossing the Mediterranean in spring and autumn. The island receives a large number of tourists annually and includes a small village that hosts a significant feral cat colony [49]. To assess the potential impact of this colony on the island’s biodiversity and to promote appropriate cat management strategies, it is necessary to know the abundance and density of cats and their use of space. Therefore, the objectives of this study were (1) to estimate the cat population and density on a small tourist island and determine whether there is a population associated with the urban area and another group of individuals living in the more natural area, and (2) to estimate the home ranges of individuals living in the urban area and those living in the scrubland area.

## 2. Materials and Methods

### 2.1. Study Area

The island of Tabarca is the only inhabited island in the Valencian Community (Spain) and has a total area of approximately 40.2 hectares. It can be divided into two distinct areas: one located at the western end where the village is located (urban area covering 10.7 ha) and a larger area covered with scrubland and abandoned *Opuntia* crops of an extension of 29.5 (Figure 1). We have considered the urban area to include both the small village within the wall and the isthmus that connects the village with the shrubland area, as it includes various types of buildings (restaurants, official buildings, warehouses, and waste management facilities). Apart from birds, the island does not have abundant terrestrial fauna, but various species of arthropods, mollusks, and vertebrates can be observed [50]. The terrestrial flora is sparse due to the dry Mediterranean climate, strong winds, and high salinity. Although it lacks trees, it has various plant species, including the endemic *Limonium furfuraceum*, which is found only in the Valencian territory [51,52]. Some of the species of fauna and flora have some category of protection.

On the island of Tabarca, a community of cats has existed for decades, possibly playing a crucial role in controlling the rat population. Two caretakers are responsible for feeding and providing veterinary care to the cats. In the past, the population reached around 300 cats, many of which were unsterilized and in poor health. Between 2016 and 2017, an institutional neutering campaign widely publicized in the media neutered at least 160 cats; however, we have no record of whether other sterilizations were carried out by volunteers before that year. Despite logistical and financial limitations, the caretakers have managed the colony using the TNR method and established feeding stations. In 2021, the Alicante City Council proposed reducing feeding stations, generating opposition from caretakers. In 2023, several sterilization campaigns were carried out, sterilizing 18 cats, but some remained unsterilized.

There are 6 feeding stations on the island (one of them provisional for relocating the cats to another point) (Figure 2). These stations are equipped with plastic bottles and hoppers containing dry food and water, sheltered in structures to prevent access by other animals and provide protection. In some areas of the island, occasionally, residents and visitors also provide water, dry food, and food scraps to the cats, and some cats have even been observed feeding from garbage containers.

### 2.2. Counting of Urban Cats

In the urban area of the island periodic surveys were conducted at an average interval of 14.5 days, between 22 October 2022, and 18 April 2023. In total, 13 surveys were carried out, mostly during the day, except on 2 occasions when nighttime surveys were conducted. During each survey, the locations of the cats detected were georeferenced, and cats were photographed to create a database that would allow the identification of individuals based on their particular color characteristics. Neutered cats were identified by the presence of ear notches, which were placed in the left ear in females and in the right ear in males. Each survey focused on visiting as many feeding points as possible as well as the intermediate areas, especially those where the caretakers knew that the cats congregated most frequently.

### 2.3. GPS Monitoring of Urban Cats

Space use in the urban area was monitored by employing i-gotU GT-120 GPS data loggers (Mobile Action Technology Inc., Taipei, Taiwan) placed on cat collars with a safety mechanism. This type of device has been successfully used in several studies on cat movements [53]. Between January and February 2023, three devices were installed on neutered males, each from a different feeding point. The collars were placed by the caretaker of the feline colony on individuals, which allowed a close approach, using wet cat food as an incentive to attract them and facilitate installation without capturing the cat. No anomalous or rejection behavior toward the device was observed in the cats.

These data loggers are lightweight and affordable and have been proven to have high success rates and a positional error comparable to more expensive devices, supporting their use in studies of free-ranging animals [54,55,56]. The devices were programmed to record a position every 4 min, 24 h a day, allowing for a maximum battery life of 7 consecutive days, after which the devices were retrieved using the same food used for installation. The average time during which these devices collected data was 164.4 h (SD = 10.1, range = 153.9–174.1).

Cat 51 from feeding point No. 3 had the GPS device from 12 January 2023 to 19 January 2023. Cat 28 from point No. 5 had it from 28 January 2023 to 4 February 2023. Finally, cat 17 from point No. 1 had it from 4 February 2023 to 11 February 2023. Appendix A shows the three individuals wearing the GPS.

### 2.4. Camera Traps in Scrubland Area

Camera traps were used to estimate the number of cats roaming in island scrublands [57,58,59,60]. Nine transects oriented north-south, thus perpendicular to the island’s east-west axis, were distributed in the scrubland area (Figure 2). Three cameras (1 Coolife H881 and 2 Sentinel Mini) were placed in the easternmost transect and were moved to the nearest transect in the westward direction every 14 days. The placement of the cameras started (27 December 2022) at the farthest end from the urban area to prevent the potential attraction of cats from the urban area toward the east. Camera trapping lasted until 2 May 2023. In each transect, a camera was located in the center of the island and the other two at points near the north and south coast. The distance between the two easternmost transects was 50 m, while the rest of the transects were spaced 100 m apart. The cameras were placed between 4 and 25 cm above the ground and oriented northward in locations that maximized the probability of capturing the entry or movement of the felines, avoiding pathways frequented by people and high-density vegetation. Approximately 50 g of tuna was used as bait [61].

In the first transect, the cameras were programmed with medium sensitivity to capture 3 consecutive photos with a 1-min delay between shots. After checking that the videos were more useful for identifying individual cats, cameras were programmed in the rest of the transects to record 10-s videos at medium sensitivity and with a 15-s delay. Cameras were always programmed to capture photos or videos 24 h a day.

In order to determine the number of cats using the feeding point closest to the scrubland area (Figure 2, number 3), a camera trap was installed within the infrastructure housing the food and drink containers. A Coolife H881 camera was used, positioned approximately two meters from the feeders. The camera was operated in two periods, during which it recorded 10-s videos at medium sensitivity 24 h a day. In the first period, between 27 and 28 February 2023, it was programmed with a 2 min delay, resulting in an excessive number of videos recorded over just one and a half days. In the second period, between 5 and 17 April 2023, it was programmed with a 15 min delay and recorded videos for 12 days.

### 2.5. Data Processing

To estimate home ranges, the minimum convex polygon (MCP) method [44,62,63] was used both in the urban and scrubland areas. MCPs were calculated in the urban zone only for cats with at least 5 records. The home range of the cats marked with GPS in the urban zone was also estimated using the kernel density estimation (KDE) method [43,64,65]. The home range area was estimated as the 95% kernel, and the core activity area as the 50% kernel. Before processing, the GPS-generated locations were filtered using the “SDLfilter” R package [66] to eliminate significant errors caused by poor signal reception or an insufficient number of satellites. The “adehabitatHR” R package [67] was used to estimate MCP and KDE. For KDE, href was used as an estimate for the bandwidth. When the home range estimated with KDE extended beyond the coastline, it was clipped using the island coastline polygon, and the areas reported were obtained after clipping. A heat map was created from all contacts obtained during the surveys to estimate the spatial variations in the intensity of cat use in the urban area. The radius used to calculate these maps (35.9 m) was derived as the mean distance between each location and the centroid of each individual, considering only individuals with at least three records. QGIS 3.22 Białowieża [68] was employed to generate the maps.

## 3. Results

### 3.1. Population Estimate

During the surveys in the urban area, 273 records were obtained, which allowed the identification of 116 individuals, consisting of 69 males (48 neutered and 21 intact) and 47 females (40 neutered and seven intact). The camera traps in the scrubland area recorded 264 images of cats, which made it possible to identify the individual in 87.8% of the cases. Using these images, eight individuals were identified (seven males: five neutered, and two intact; and one neutered female) (Appendix A) that were different from those in the urban area and were never detected there. The sex ratio was more balanced in the urban area, where the proportion of males was 59.5%, while in the scrubland area, 87.5% of the individuals were males.

In the westernmost scrubland camera trap transect, which was closest to the urban area and feeding point No. 3 (Figure 2), four additional individuals were detected that had been recorded previously during the surveys in the urban area but were not detected in the rest of the camera trap transects. Therefore, the cat population on the island has been estimated at 124 individuals (76 males and 48 females), with a total density of 308 cats/km^2^ (1084 cats/km^2^ in the urban area and 27 cats/km^2^ in the rural area). 

At the start of the study, 94 cats were neutered (75.8%; 41 females and 53 males), and 30 were intact (24.2%; 7 females and 23 males). The caretakers of the cat colony, with the support of the Alicante City Council, conducted several capture sessions for neutering between 2, 16, and 29 March 2023. During these sessions, 18 individuals were neutered (16 from the urban area —13 males and three pregnant females—and two males from the scrubland area), increasing the neutering rate to 90.3% (89.5% of males and 91.7% of females). After these campaigns, 12 individuals remained unneutered, all from the urban area (eight males and four females). It was unknown whether some of the unneutered four females could be pregnant. During the study, the deaths of six individuals registered in the urban area were recorded (two neutered females, three neutered males, and one unneutered male), resulting in a population of 118 cats (72 males and 46 females) at the end of the study period.

### 3.2. Home Range 

The map of the intensity of use of the urban area (Figure 3) shows that the areas near feeding points are among the most frequented by individuals, although the intensity of use varied between them and was particularly high at points 2, 3, and 4. Outside feeding points, there are other areas with high intensity of use, located at the island harbor, at the restaurants before entering the village, and at Plaza Grande, located in the center of the urban area, where cats also visit the restaurants.

The home ranges estimated for the cats in the urban area are shown in Table 1 and Figure 4. The areas estimated from the sightings during the surveys covered a wide range (0.003–1.29 ha) and were not correlated with the number of records of the individuals (r = 0.059). The two individuals with the largest home ranges were males that were not neutered, although in 2 March 2023, one of them was neutered. The average home range estimated by this method yielded 0.38 ha (SD = 0.38) and was very similar for males (0.39 ha) and females (0.33 ha), although data from only two females were included.

The home ranges (KDE 95%) and core areas (KDE 50%) of the three neutered male cats equipped with GPS in the urban area are shown in Figure 4. The areas were very similar among the three individuals (Table 1) and were, on average, 1.25 ha (SD = 0.047) for the KDE 95% and 0.24 ha (SD = 0.051) for the KDE 50%. The average home range was remarkably similar to the maximum area estimated from the survey locations. There was no overlap between the home ranges of any individual, even though cats 28 and 17 moved in very close areas. The core area is located around the feeding points for cats 28 and 51, while for cat 17, it does not encompass the feeding point.

The home ranges of the eight individuals identified in the scrubland area were all larger than those estimated in the urban area, ranging from just under 3 ha to 13.9 ha (Table 2), with an average of 9.53 ha (SD = 3.65). The only female in this group was the individual with the largest home range. The individuals detected in the scrubland area covered nearly the entire available space up to the point on the island farthest from the urban area, and their home ranges overlapped extensively (Figure 5).

The camera located at the feeding point near the scrubland area recorded 715 cat visits, in which the individual could be identified in 95.94% of them. Thirty individuals were identified (22 cats from the urban area and eight cats from the scrubland area). This feeding point was used by cats that moved through the eastern half of the urban area (Figure 5) and by all cats from the scrubland area. At this point, 56.6% (17/30) of the individuals were unneutered before the sterilization campaign, a percentage that decreased to 50% (6/12) after these campaigns.

## 4. Discussion

Our results show that Tabarca Island has one of the highest cat densities reported on islands (overall 308 cats/km^2^; urban zone 1084 cats/km^2^; scrubland zone 27 cats/km^2^). Due to its very small size, it has been possible to accurately estimate the density and total population of the two major habitat types present. Very few studies have estimated the cat density on islands by considering the entire area. Among the studies reviewed by Nogales et al. [69], apparently, only the study by Fitzgerald and Veitch [70] conducted on the small island of Herekopare (30 ha) in New Zealand estimated the total island population by capturing the entire population, and it reported a density of 120 cats/km^2^. Most studies on cat density in islands have been conducted on much larger islands, where cat density has been estimated by sampling specific areas within the islands. However, as shown by the difference in density between the two habitats in Tabarca, it is possible that densities estimated by sampling specific habitats may not be representative of total density. Many studies that have not covered the entire island report much lower densities than those found in Tabarca. For example, in the Ecological Park of Funchal on Madeira Island, the density was 1.4 cats/km^2^ [71], and in an area within the National Park of Réunion Island, the density was 0.25 cats/km^2^ [72]. In certain areas of Guadalcanal and Kolombangara (Solomon Islands), densities ranged between 0.31 and 2.65 cats/km^2^ and 0.65 cats/km^2^, respectively [73]. In their review of 27 islands where cats have been eradicated, Nogales et al. [69] found a maximum density of 243.3 cats/km^2^ on the small Cousine Island (30 ha) in Seychelles, although most islands had much lower densities (overall average 33.41 cats/km^2^). In some areas of Kangaroo Island, South Australia, the average density was 0.37 cats/km^2^ (0.06–3.27 cats/km^2^) [74]. On a peninsula at the northern tip of Auckland Island, south of New Zealand, a density between 0.7–1 cats/km^2^ was estimated [75].

In urban areas, the density of cats has been found to be highly variable, although values reported are generally lower than those recorded in Tabarca. In Bristol, United Kingdom, density has been estimated as 229 domestic cats/km^2^ [76]; in Guelph and Windsor, Canada, 0.494 and 13.3 stray cats/km^2^ [77,78]. Sims et al. [16] estimated a density of 417 domestic cats/km^2^ (132–1580 cats/km^2^) in urban areas of Great Britain, and McDonald and Skillings [79] estimated between 1.9 and 57 cats/km^2^, with an average of 9.3 cats/km^2^, in five urban areas of the United Kingdom. In South Korea, across six selected districts, the average density was 186.8 (132–268) stray cats/km^2^ [80], and in a historic neighborhood of Puerto Rico, it was 360 stray cats/km^2^ [81]. On Corvo Island, densities varied between the uninhabited area (3.6; 2.5–5.4 cats/km^2^) and the inhabited area (73.4; 58.1–92.7 domestic cats/km^2^) [82]. Densely populated areas tend to have more stray cats [13,18,19,76,77,79,81], as reflected in the urban area of Tabarca, which may be a source of cats in the scrubland area. In summary, Tabarca has one of the highest cat densities reported in the literature, whether considering the island as a whole or just the urban area.

The sterilization rate at the end of the study was 90.3%. Before the sterilization campaigns, the percentages were similar in the urban and scrubland areas at 75.86% (88/116) and 75% (6/8), respectively. This suggests that the high number of unsterilized cats of both sexes in the urban area could originate new litter and that this area might act as a source of individuals for the scrubland area. Although there is controversy in the scientific literature regarding the effectiveness of the TNR method for controlling cat colonies [31,35,83,84,85,86], the total sterilization rate is close to the 100% needed for the effectiveness of TNR according to Levy et al. [87] and Foley et al. [88]. Gunther et al. [85] mention that more than 70% of sterilization reduces the population by approximately 7% annually. Therefore, if this prediction holds true in Tabarca, it would take about 10 years for the cat population to be halved, and in that case, its density would still be within the high range of densities of the islands reviewed by [69]. 

The higher proportion of males in the scrubland area, where only one individual was female, suggests a greater tendency for males to move to this area. This occurs even if the males are neutered since most of the males in the scrubland area were neutered before the start of the study. Various studies have shown that males have, on average, larger home ranges than females, both during and outside the breeding season [89,90,91,92,93,94,95], which could increase the likelihood of males from the urban area moving to the scrubland area and establishing their home range there compared to females. Cats from the scrubland area were not detected in the urban area, except at feeding point No. 3, which is located at the border. On the contrary, four cats from the urban area (three males and one female) were detected on the scrubland camera traps closest to the village, supporting the idea that there is a main flow of individuals from the urban area to the scrubland area.

During the surveys in the urban area, it was difficult to obtain sufficient observations of individual cats to reliably estimate their home ranges. Consequently, only 12 cats (10.3% of the population), most of them males, reached the threshold of five records required to calculate the MCP. The home ranges estimated in the urban area from these surveys were very small, and in the only individual where it was also estimated using KDE, it was seven times larger than the estimate with MCP. This strongly suggests that the spatial distribution of observations during the surveys underestimates the home range. It is possible that at least part of the difference is due to the fact that most of the contacts obtained during the surveys were collected during the day (97.1%), whereas the data obtained from GPS-tagged cats encompassed the entire daily cycle. It is also likely that most of the contacts obtained in these surveys occur in places where cats are more detectable or seek human proximity, such as feeding points or restaurants.

The home ranges of all cats in the scrubland area, estimated using MCP, were larger than those of cats moving in the urban area, estimated using GPS, even though estimates obtained by MCP from camera trap data are likely to underestimate the home range because the locations are limited by the position of the cameras. Choeur et al. [91] found a similar result on Réunion Island but compared unowned cats with owned cats, where the home range of the former was nearly four times larger than the latter. In our study, all cats were unowned, and the difference lies in the area they inhabit, such that in Tabarca, urban area cats behave like owned cats, as the village is very small with six feeding points and numerous restaurants where they often feed on food scraps, similar to observations on Réunion Island [91]. Cats in the scrubland area behave like unowned cats, and their home range is about seven times larger than that of urban area cats.

Compared to other studies, cat home ranges are smaller in Tabarca, largely due to the small size of the island (40.2 ha), and much smaller than the home ranges reported in numerous studies, which can be tens or hundreds of hectares. For example, Bengsen et al. [57] reported an average home range of 511 ha on Kangaroo Island. Goltz et al. [96] observed in Mauna Kea, Hawaii, that males had larger home ranges than females (1418–2050 ha for males and 772 ha for females). In New Zealand, Nottingham et al. [95] also recorded larger home ranges for males (22.1–3232 ha) than for females (9.6–2078 ha). On Rota Island in the Mariana Islands, Leo et al. [94] found that males had home ranges of approximately 132 ha, while females occupied areas of approximately 22 ha. On Socorro Island, in the Revillagigedo Archipelago, Ortiz-Alcaraz et al. [92] recorded average home ranges of 219.1 ha for males and 118.86 ha for females. In Hawaii, Smucker et al. [93] estimated that males occupied areas of 574 ± 273 ha, compared to 223 ± 44 ha for females. The extent of these home ranges exceeds by far the size of small islands, such as Tabarca, Herekopare, or Cousin Island, which range between 30 and 40 ha. This small area relative to the potential home range that the species can achieve likely influences their behavior, forcing smaller home ranges and reducing the differences between males and females. In our study, we did not obtain enough information to compare the home ranges of males and females, but the results do not support a significant difference between the sexes in Tabarca. Conversely, on the small Japanese island of Ainoshima (125 ha), which is three times larger than Tabarca and has a similar population (130 cats), the home range, estimated using a method similar to one we used in Tabarca (MCP from visual observations obtained during surveys), showed average values of 0.78 ha (non-estrous season) and 1.45 ha (estrous season) for males and 0.44 ha and 0.61 ha in those same seasons for females [29].

The areas near feeding points are the most frequented and show a higher intensity of use, which is in line with the findings of [61]. Additionally, some areas without feeding points also show a high intensity of use due to other food sources provided by residents, tourists, and local restaurants, which also contributes to maintaining feline populations and affects home ranges [91]. This was evidenced by the presence of containers with human food scraps in the study area and the observation of foraging behavior in dumpsters [97,98]. The home ranges of the GPS-collared cats did not overlap, as each individual was associated with a feeding point and stayed very close to it despite the short distances between them. At each feeding point, the individuals that used it had overlapping ranges, as inferred from the locations of their observations during surveys. In the scrubland area, the home ranges of all individuals overlapped extensively, and no territorial behaviors were observed. Several cats were even seen together during their movements through this area.

Domestic cats are generalist predators that capture a wide variety of wild prey [7,26], even when provided with food by their owners or caretakers [99]. The most significant impacts are observed on islands [8,100,101,102], where small mammals [76,103,104], birds [26,97], reptiles [104,105], invertebrates, and amphibians [2,7,106] are particularly vulnerable. The island of Tabarca, included in the network of Special Protection Areas for birds, is a breeding area for various seabird species, as well as passerines. It used to be a breeding ground for the Kentish plover (*Charadrius alexandrinus*), which has recently disappeared as a breeder [107]. Additionally, Tabarca is a stopover point for migratory birds [108] that frequently arrive on the island under poor physical conditions [109]. Cats can have a potential negative impact on the island’s fauna, as migratory birds and fledglings that spend a lot of time in low vegetation, characteristic of the entire island, are especially vulnerable to predation [110]. Furthermore, the presence of cats can have other effects, such as reduced reproductive success [22], by causing nest abandonment and increasing predation risk [111,112]. They can also cause avoidance of suitable habitats [113], increase vigilance, and decrease feeding rates [114] in both adults and chicks [111]. All these effects are intensified when cat density is high, and cats hunting for birds, mammals, and reptiles in Tabarca have been reported previously or observed by the authors. Therefore, management measures should be adopted to reduce the number of cats on the island. The legislation in Spain prohibits culling as a measure of controlling cat colony size, but not their relocation [115]. The legal conditions that allow for the relocation of community cats include negative impacts on biodiversity in protected natural areas and Natura 2000 sites as well as on protected wildlife, which would be applicable in this case. Since relocating a large group of cats, like the one in Tabarca, can pose a significant logistical challenge, we propose a solution in several steps that would be easier to implement. The greatest impact on biodiversity is expected in the shrubland area; thus, we believe it is more urgent to address the problem in this zone. Therefore, we propose that the environmental and local authorities declare the shrubland area of the island as a cat exclusion zone [43]. This would involve monitoring the area with camera traps, capturing newly detected individuals, and transferring them to an authorized site. Additionally, feeding point No. 3, located on the edge of the urban area, should be moved further inside the urban zone away from the shrubland to reduce the likelihood of urban cats using this point to move into the shrubland area. Finally, given the difficulties of neutering the entire population solely through the efforts of volunteers, it is recommended that local institutions engage specialized companies for the complete sterilization of the entire urban cat population.

## 5. Conclusions

The island of Tabarca has one of the highest densities of cats reported in the literature, which we have estimated to be 308 cats/km^2^ for the entire island, reaching 1084 cats/km^2^ in the small urban area. Our data support that this urban area acts as a source of cats for the scrubland zone, where we estimated a density of 27 cats/km^2^. The home ranges of cats in Tabarca are very small compared to those estimated on larger islands, most likely due to the small size of the study island. Additionally, there are differences in the size of the home ranges of cats depending on the zone they occupy. In the urban area, cats have a much smaller home range than those in the scrubland zone. In both areas, there is extensive overlap of home ranges among different individuals, and no territorial behavior is observed, likely due to the high percentage of sterilized cats. Cats have been observed hunting birds, mammals, and reptiles in Tabarca. Given the high density of cats, measures should be taken to reduce their population, especially considering that the island is included in the Natura 2000 Network as a Special Protection Area (SPA).

## Figures and Tables

**Figure 1 animals-14-02288-f001:**
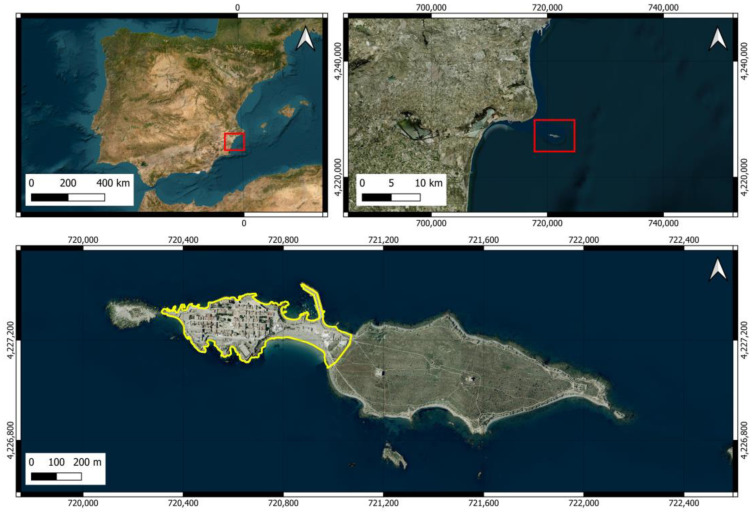
Location of Tabarca Island. The area marked by the red rectangle in an image outlines the area that appears enlarged in the next one. The yellow line delineates the urban area of the island.

**Figure 2 animals-14-02288-f002:**
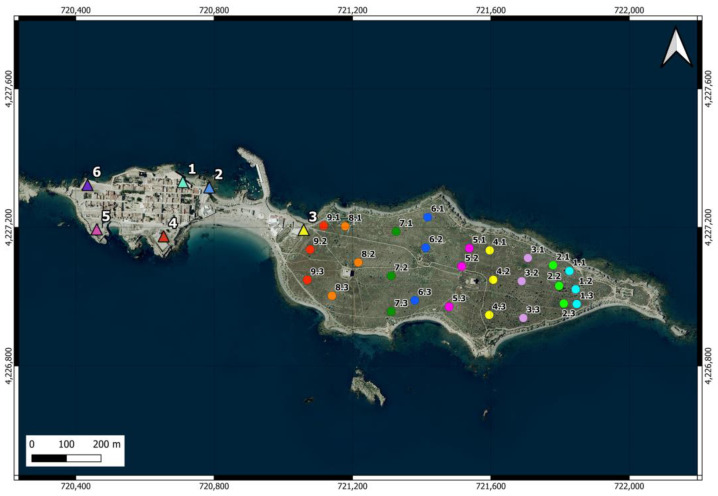
Location of the feeding points of the cat colony (triangles in the urban area) and the 27 camera trap points distributed across 9 north-south oriented transects (circles in the scrubland area) on Tabarca Island. In the latter, the first number identifies the transect, and the second indicates the position. The points within the same transect, where the cameras operated simultaneously, are shown in the same color. The cameras were moved from east to west.

**Figure 3 animals-14-02288-f003:**
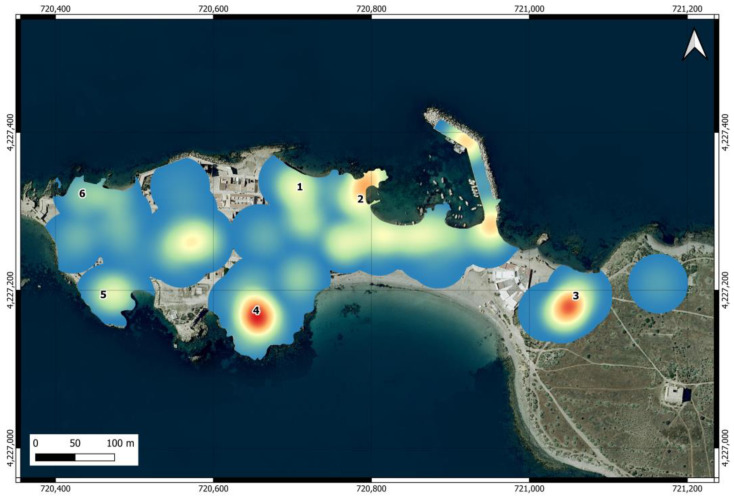
Heat map of the intensity of use of the urban area of Tabarca by cats. The numbers indicate the locations of official feeding points.

**Figure 4 animals-14-02288-f004:**
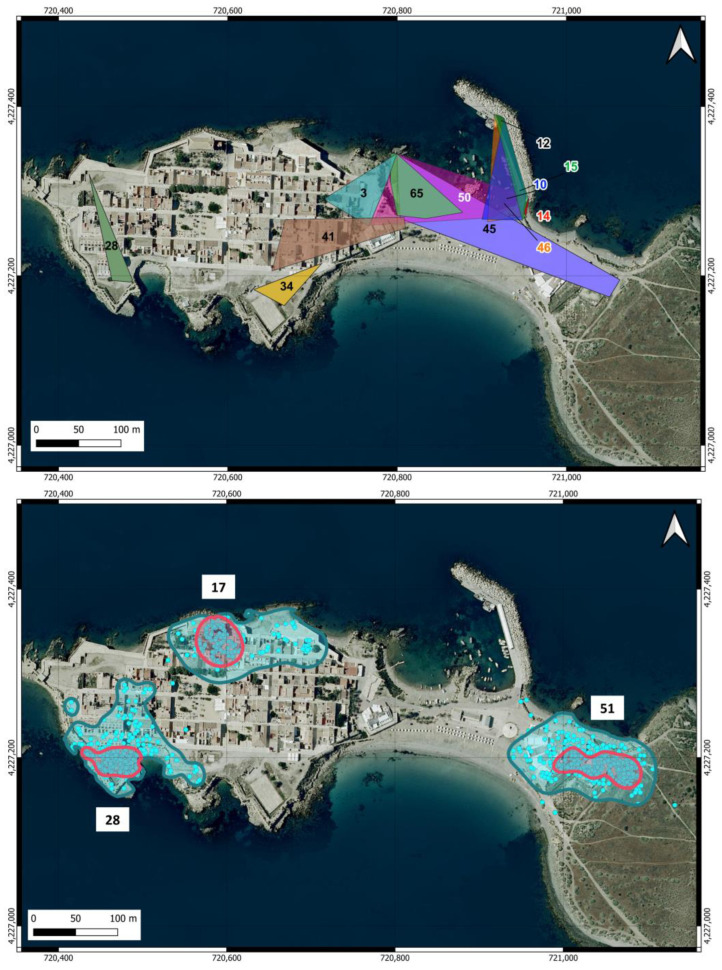
Home ranges of cats in the urban area of Tabarca estimated using two methods. Upper map: MCP calculated from records of cats with at least five locations obtained during the surveys. Lower map: The home range was estimated as KDE 95% (blue line), and core areas were estimated as KDE 50% (pink line) for the three neutered male cats equipped with GPS. The blue dots show the locations used for the calculations. The numbers in both maps indicate the cats listed in Table 1.

**Figure 5 animals-14-02288-f005:**
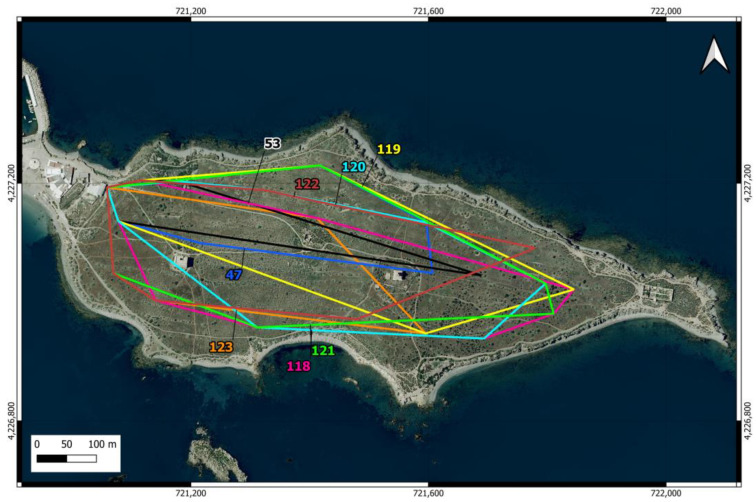
Home ranges of cats in the scrubland area of Tabarca Island, estimated using MCP from locations obtained through camera trapping. Cats are identified by the same number as in Table 2.

**Table 1 animals-14-02288-t001:** Home range (ha) estimated using MCP (Minimum Convex Polygon) from the locations obtained during the surveys (MCP Survey, only cats detected in at least five surveys) and kernels (KDE 95% and KDE 50%) for individuals marked with GPS. For these individuals, Days show the number of days that the GPS recorded their position. MCP GPS is estimated using the GPS positions. N locs survey: number of locations obtained during the survey. N locs GPS: number of positions obtained by the GPS after filtering. Neutered: identifies cats that were neutered before the start of the study. An individual neutered during the study is marked with an asterisk.

Cat ID	Days	Sex	Neutered	N Locs Survey	MCP Survey	N Locs GPS	KDE 50%	KDE 95%	MCP GPS
45	-	M	N *	6	1.29				
50	-	M	N	5	0.90				
41	-	F	Y	6	0.53				
46	-	M	Y	5	0.37				
10	-	M	Y	5	0.35				
65	-	M	N	6	0.34				
3	-	M	Y	6	0.29				
28	7.0	M	Y	5	0.16	443	0.18	1.20	1.79
15	-	M	Y	8	0.14				
34	-	F	Y	5	0.13				
12	-	M	Y	5	0.04				
14	-	M	Y	5	0.003				
51	6.4	M	Y			541	0.28	1.29	1.57
17	6.9	M	Y			181	0.25	1.27	0.87

**Table 2 animals-14-02288-t002:** Home ranges of the cats identified in the scrubland area, estimated as the MCP of identification in camera traps. Neutered: Individuals that were neutered before the start of the study. The individuals who were sterilized during the study period are marked with an asterisk.

Cat ID	Sex	Neutered	N Locs Cameras	Home Range (ha)
121	F	Y	12	13.91
120	M	Y	16	12.01
118	M	Y	14	11.84
119	M	Y	13	10.98
122	M	N *	10	10.80
123	M	N *	7	7.55
47	M	Y	10	6.21
53	M	Y	5	2.91

## Data Availability

The raw data supporting the conclusions of this article will be made available by the authors upon request.

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
