# Peer review of "Density and Home Range of Cats in a Small Inhabited Mediterranean Island"

_animals, 2024, doi:10.3390/ani14162288_

Round 1

Reviewer 1 Report

Comments and Suggestions for Authors

I do not do quantitative research, so my analysis of the methodology and results is fairly superficial.  I am also unfamiliar with some of the data-presentation methods used in this article, and I found them informative and well used.  The article was well-written, though some paragraphs seemed rather haphazardly organized (e.g., lines 79-87), and I  came upon several typos (e.g., lines 150, 330).

If there is an academic convention of allowing a parenthetical citation stand in for a grammatical subject or object in a sentence, I am unaware of it.  I found this to be stylistically unfriendly to a reader.

In terms of content, my only suggestion is that the authors clarify their references to "additional management measures," or "complementary measures."  In the concluding lines of the article, the authors vaguely assert a moral imperative based on their data: "Given the high density of cats, measures should be taken to reduce their population."  The evasiveness of this assertion comes from the abstraction of "measures" coupled with the passive verb  (should be taken) that makes it  anybody's guess who should assume responsibility for reducing the cat population density with these mysteriously abstract measures.  In line 68, the authors suggest that these methods might be more precisely understood as "extraction and lethal control," and there is also a suggestion of concern that the methods be "socially acceptable" (79).

Immediately preceding the conclusion is information about the cats' predatory behaviors, particularly on nesting or migrating birds.  I would like to see the authors be more explicit about the ecological framework that values the functions of these birds in a biodiverse planet, making intervention urgent on their behalf (10 years is too long to wait). 

The authors have indicated what every cat owner knows--that provisioning cats doesn't necessarily decrease their predatory behavior.  Yet the data show that cat populations are most dense where human populations are most dense, and the authors note the presence of restaurants and other provisioning sources to explain this correlation.  The problem, then, is not just cats--it's the human-cat relationship.  With large predators such as bears, whose predation may include humans, management policies include education about the predator's behavior and information on how humans can change their behaviors to facilitate coexistence.  Do the "measures" supported by this study include education and possible changes in human behavior as well as the euphemistically referenced extraction and lethal control?  And if education is part of these measures, does that include ecological education about the functions of cats, birds and humans on this island environment? 

At one point the authors categorize feline behavior types as "owned" and "unowned."  This might be the point for considering the contributions of human behavior to this island ecology.

Reviewer 2 Report

Comments and Suggestions for Authors

8 – remove comma – remove ‘had’ from have had a significant

14 – ‘value’ not values

 18 – maybe give the percentage of queens sterilized here?

32- is this a combined male and female rate or female only?

51 – maybe give the name of the [23] citation here since it’s at the beginning of the sentence

56 – cause instead of causes

59 – Are you trying to emphasize that even a small human population can beget a large

70 – change sentence to : However, social factors may hinder its implementation on inhabited islands

97 – change touristic to tourist

120 – you might want to clarify that the sterilization campaign began in 2016, unless youre saying that the target population reduction was achieved in 2016. It’s unclear here.

These are very high quality maps of the islands. Excellent job presenting the visuals here.

204-the distinction here between mcp and kde is clear, but later on (specifically around the 366 paragraph) you use mcp vs gps as the descriptor. I would have a clear set of  group names and stick with them throughout for clarity.

215- just clarifying, a record here is an unique observation of a cat?

Very clear description of the range methods.

295- I’m not clear on the particular 95% range method here, but is there a problem that the range of an animal on the island will never include the beach? Does it shrink away 5% away or will the formula weight the home range right up to the waters edge?

363 – I’m not sure I see the argument here that the movement of new cats in the system will be from urban to rural. If they both have similar sterilization percentages but have very different densities. The 4 cats that were ‘hybrid’ here were not strictly moving from a urban to a rural areas, but just had a home range that straddled the urban/rural edge zone correct?

Broad question on categorization:

My apologies if I missed it, but what is your demarcation system for calling an area urban or rural? If 10.7 ha are urban and it has such a high cat density (1084/km2) then a relatively minor recategorization of an area as rural (or the reverse) would drastically throw off your island wide estimate.

For example, the area between the #2 and #3 feeding stations. I don’t see a lot of buildings on the overhead view, but I think that figure 3 is calling it ‘urban’. There obviously has to be a cutoff somewhere between your zones, but surely it doesn’t crash from 1k cats/ km2 to 30 cats /km2 on a dividing line. If you were to classify an ‘edge zone’ with possibly a median cat density, how would that influence your measurements? Does that actually bear any semblance to reality or is there really that stark of a difference?                                                                                                                        

Comments on the Quality of English Language

Very minor edits, great job overall
